# Divergent Conceptualization of Embodied Emotions in the English and Chinese Languages

**DOI:** 10.3390/brainsci12070911

**Published:** 2022-07-13

**Authors:** Pin Zhou, Hugo Critchley, Yoko Nagai, Chao Wang

**Affiliations:** 1College of Foreign Languages, Shanghai Maritime University, Shanghai 201306, China; wangchao0022@stu.shmtu.edu.cn; 2Sackler Centre for Consciousness Science, University of Sussex, Brighton BN1 9RN, UK; h.critchley@bsms.ac.uk; 3BSMS Department of Neuroscience, Brighton and Sussex Medical School, University of Sussex, Brighton BN1 9 RX, UK; y.nagai@bsms.ac.uk

**Keywords:** emotion, conceptualization, interoception, afferent, efferent

## Abstract

Traditional cognitive linguistic theories acknowledge that human emotions are embodied, yet they fail to distinguish the dimensions that reflect the direction of neural signaling between the brain and body. Differences exist across languages and cultures in whether embodied emotions are conceptualized as *afferent* (feelings from the body) or *efferent* (enacted through the body). This important distinction has been neglected in academic discourse, arguably as a consequence of the ‘lexical approach’, and the dominance within the affective psychology of the cognitive and semantic models that overlook the role of interoception as an essential component of affective experience. Empirical and theoretical advances in human neuroscience are driving a reappraisal of the relationships between the mind, brain and body, with particular relevance to emotions. Allostatic (predictive) control of the internal bodily states is considered fundamental to the experience of emotions enacted through interoceptive sensory feelings and through the evoked physiological and physical actions mediated through efferent neural pathways. Embodied emotion concepts encompass these categorized outcomes of bidirectional brain–body interactions yet can be differentiated further into afferent or interoceptive and efferent or autonomic processes. Between languages, a comparison of emotion words indicates the dominance of afferent or interoceptive processes in how embodied emotions are conceptualized in Chinese, while efferent or autonomic processes feature more commonly in English. Correspondingly, in linguistic expressions of emotion, Chinese-speaking people are biased toward being more receptive, reflective, and adaptive, whereas native English speakers may tend to be more reactive, proactive, and interactive. Arguably, these distinct conceptual models of emotions may shape the perceived divergent values and ‘national character’ of Chinese- and English-speaking cultures.

## 1. Reexamination of the Embodied Emotional Language within the Framework of Interoceptive Cognitive Theory

References to the body are one feature shared across languages, particularly when describing the mental processes of emotion, reflecting the embodiment of an emotional experience. Through the investigation of an impressive number and variety of languages, anthropological linguists claim that ‘emotions can be linguistically represented via literal somatic sensations (e.g., *she blushed*) or by body-part phrases referring to both literal and imaginary processes taking place inside or with the body (e.g., *his hair stood on his head, his heart sank, it just makes my blood boil*)’ [1] (p. 148). Beyond this general tendency to link emotional states to bodily states, languages nevertheless vary dramatically in how they embody emotion concepts. Some researchers generalize three broad conceptualizations of the mind and emotion categories globally. The first is abdominocentrism (i.e., the mind, including feeling, thinking, and knowing, is conceptualized as being located in or around the abdomen region (e.g., in the belly, in the liver, or in the kidneys)). Southern Asia, Polynesia, and other disparate cultures, including Basque culture, are the prototypical abdominocentric cultures. Next is cardiocentrism (i.e., the mind and emotions are located in the heart region). Cardiocentrism is the traditional view of China, Korea, and Japan, with their sharing similar philosophical and medical cultural models of holistic, heart-centering conceptualizations. Lastly, there is cerebrocentrism (i.e., the mind is believed to be seated in the head or the brain region). This conception of the mind is mainly held by Greek-based West Asian, European, and North African cultures, with the prototypical examples being the major Indo-European languages [1,2,3].

While some academics claim that many non-Western languages do not appear to differentiate between emotions and bodily sensations to the same extent that Western languages do [1], others propose that Western and non-Western languages are located at the two poles of a continuum between bodily transparency and cognitive granularity in the conceptualization of emotions. For example, Chinese is comparatively higher in bodily transparency but lower in cognitive granularity, while Western languages manifest the opposite relationship [4].

Nonetheless, the term ‘embodied emotion’ remains a broad concept, with the two dimensions of afferent (i.e., body to brain) versus efferent (i.e., brain to body) flow of information undifferentiated. Arguably, the contribution of the body to emotions should be differentiated along these complementary routes of brain–body interaction (i.e., the ‘bottom-up’ interoceptive sensory signals passing along the afferent pathways and the top-down autonomic nervous drive to change the internal bodily state transmitted along the efferent pathway). A more granular understanding of embodied cognitive processes and the brain–body relationship will be enriched by consideration of this distinction.

Within modern cognitive psychology and neuroscience, the embodiment of mental processes has increasingly been recognized (i.e., ‘brains are in bodies’) [5]. Moreover, a shift in perspective challenges the notion that brain is the ‘master’ or ‘commander-in-chief’ of the body, as generally assumed by standard philosophy of the mind and cognitive science. Instead, the brain is viewed as the ‘servant’ of the body, with the primary function of maintaining homeostasis [6,7]. This revolutionary framing of the brain–body relationship has broad and potentially radical implications for how we understand mental processes and the intricate, multi-dimensional relationship between language, culture, the mind, emotion, the brain, and the body [4]. Fundamentally, the brain is proposed to support the whole body by predicting or inferring the energy needed to manage future circumstances effectively, grounded by the imperative of maintaining long-term homeostatic equilibrium through responses informed by interoceptive input from the internal organs and weighed against expectations (beliefs) and previous experience. This energy-budgeting function is the essence of allostasis, in which the brain automatically predicts, prepares for, and calculates the energy to be expended by the body before an event actually happens. Here, biological resources that maintain the homeostatic order of the body, such as the supply of water, salt, and glucose, have primacy [5,8,9]. Thus, within the framework of control theory or cybernetics, the brain works to keep the equilibrium of essential variables in the body through minimizing or reducing the free energy or surprise arising from prediction errors [10,11,12,13,14]. The allostatic notion of brain–body interaction or an ‘inference coding system’ therefore works in a top-down manner [5].

In line with this updated view on the brain–body relationship, recognizing the interplay of top-down expectations with bottom-up sensory signaling from the body, the embodiment of emotion concepts across cultures and languages (e.g., Chinese and English) merits reexamination. The present paper will first review how the embodiment of emotions is mapped through afferent (interoceptive) and efferent (autonomic) channels. Second, the words, idioms, and their usage examples (listed within relevant thesauri and on-line dictionaries) used to describe so-called ‘basic’ emotions such as *fear*, *anger*, *sadness*, and *joy* will be compared between the Chinese and English languages to explore how embodiment is used in the conceptualization of emotions. Finally, the distinct ways in which the Chinese and English languages embody emotional concepts will be examined as an account of how perceptions about the stereotypical differences lying in their cultural values and ‘national personalities’ may be constructed regarding Chinese- and English-speaking people from their respective conceptions of the body [15,16].

## 2. The Distinct Embodying Processes in Generating Emotion Concepts

Rather than drawing on the ‘lexical approach’ to speculate the mental mechanism of emotion concepts as commonly practiced in the field of cognitive linguistics [17,18,19,20,21], the embodiment of emotion concepts, grounded by modern embodied cognitive science, particularly interoceptive neuroscience, can be broadly parsed along two neural axes. The first is afferent body-to-brain ‘interoception’, encompassing the signaling and representation of changes in the physiological state translated within the brain into feeling states of subjective emotional experience and awareness. The second is the efferent route from the brain to the body, whereby physical actions and autonomically mediated physiological changes are engendered in the body by emotions. Thus, the neural signals of these two embodying processes involved in emotional experiences flow in the opposite direction.

The term ‘interoception’ refers to the neural signaling and representation (both unconscious and conscious) of signals pertaining to the internal state of the body, encompassing sensations of pain, temperature, bloating, itch, hunger, thirst, muscle burn, joint ache, sensual touch, flushing, visceral urgency, and nausea that originate from the activation of receptors within the visceral tissues of the body, including nociceptors, thermoreceptors, osmoreceptors, and metaboreceptors [22].

The afferent fibers then transmit interoceptive signals through the spinal cord and via the cranial nerves (mostly the vagus nerve) to the brainstem first, where they interact extensively with efferent autonomic centers that can elicit a nearly instantaneous physiological response to implement homeostatic autonomic control [23]. As the interoceptive signals ascend in the brain, they are projected to multiple nuclei within the periaqueductal gray (PAG), the parabrachial nucleus (PBN), the nucleus of the solitary tract (NTS), the thalamus (notably the ventromedial nucleus), and the insular cortex (IC) [24] (p. 4) and then are processed and translated into subjective motivational and affective (pleasant or unpleasant) feelings that motivate individuals (consciously or not) to approach or avoid [23,25] so as to guide and maintain homeostatic equilibrium. This essential role of interoception in motivating actions to ensure bodily homeostasis is enabled by ‘predictive processing’ [8,9,22,26,27]. The predictive processing framework argues that the brain infers (probabilistically or on an ‘approximate Bayesian’ basis) the likely cause of changes in sensory information by testing the perceived sensory data against its own predictions (‘beliefs’ of ‘priors’) [28]. Although commonly applied to sensory information about the external world, predictive processing is also applied to the internal physiological state, wherein predictions about interoceptive information are generated from an internal model of the homeostatic state of the body. When the sensory data do not match the prediction, this generates prediction errors (sensory surprise). The brain seeks to minimize prediction error by modifying its internal model (through learning) or modifying the source of the sensory data through actions (which can be autonomic responses).

Thus, the autonomic efferent drive maintains the homeostasis of an organism not only by generating contextual responses to changing incoming interoceptive bodily cues but also acting allostatically in anticipation of physiological challenges signaled by external motivational and emotional cues. Predictive coding is argued to be one instantiation of the more general free energy principle [10]. Minimizing the free energy of the internal state in order to avoid sensory surprise is equivalent to minimizing the interoceptive prediction error. Active inference (i.e., targeted action on the external (via behavior) or internal environment (via autonomic responses)) will diminish uncertainty and facilitate the reduction of free energy [29]. The actions or autonomic responses equate to sensory predictions. Ultimately, the structural and dynamic physiological integrity of a person or organism is ensured by maintenance of the physiological state within set bounds by engaging in integrated interoceptive control and autonomic actions that resist the tendency toward disorder evoked by ever-changing external conditions [10,30,31].

The ascending interoceptive sensations and motivations interact with descending predictions that manifest as autonomic responses to produce ‘homeostatic emotions’ [23], or ‘background emotions’ [32], maintaining a ‘core affect’ [33,34]. Emotions can thus be defined as the combination of bodily sensation (feelings), motivational drive, and autonomic sequelae [27] that emerges from interoceptive embodiment and integration with concurrent exteroceptive sensory information [13]. Here, interoception is the physiological substrate for feeling states (i.e., the subjective sensations of emotions), whose signals pass along the afferent, ascending neural fibers from the body to the brain, while the autonomic and motor responses responsible for the physiological and behavioral expression of emotions are conveyed through the efferent, descending neural routes from the brain to the body. Interestingly, the dominant way in which embodiment is operationalized in the conceptualization of emotions appears to diverge in the Chinese and English languages along these afferent vs. efferent pathways. Here, we will compare the embodiment of four ‘basic’ emotions—*fear*, *anger*, *sadness*, and *joy*—in these two languages.

## 3. Divergent Embodiment Underlying the ‘Basic’ Emotions in Chinese and English

Although abundant embodied emotion words (i.e., words, phrases, and idioms (i.e., verbal expressions describing emotion concepts via bodily parts, physiological changes, and reactions)) can be found in both Chinese and English, they are characterized by a reversed direction in the flow of neural signals between the brain and the body. Notably, the afferent route appears much more frequently in the Chinese conceptualization and verbalization of emotions, while the efferent pathway in emotion processing is more commonly highlighted in English. In other words, the sensory processes governed by the interoceptive systems underpinning subjective awareness of emotions are given prominence in Chinese descriptors of the affective state, while action-related and autonomic bodily responses expressing emotions preoccupy English emotion concepts. To explain this point further, the English words (synonyms) labeling the four so-called ‘basic’ emotions, namely, *fear*, *anger*, *sadness*, and *joy*, were collected from *Roget’s 21st Century Thesaurus* (the 7th edition) [35] as well as from online dictionaries such as https://www.collinsdictionary.com/us/dictionary/english-thesaurus/fear (accessed on 3 January 2022) [36] and https://www.sketchengine.eu/skell (accessed on 24 June 2022) [37], from which the emotion words with reference to body parts, physical states, and physiological reactions were sorted out and illustrated with examples. As for the collection of Chinese data, we first searched the keywords ‘the idioms that are associated with visceral organs’ and ‘the words or idioms describing *anger*, *fear*, *sadness*, or *joy*’ on www.baidu.com (accessed on 4 January 2022) [38], the most popular and frequently used search engine in China, from which several relevant websites were available such as https://wenku.baidu.com/view/1bac147ee2bd960590c67796.html (accessed on 6 January 2022) [39], http://zhidao.baidu.com (accessed on 5 January 2022) [40], http://xh.5156edu.com (accessed on 5 January 2022) [41], http://chengyu.t086.com (accessed on 4 January 2022) [42], and http://www.hydcd.com (accessed on 4 January 2022) [43], wherein we singled out the embodied emotion words and idioms and subsumed them to the four basic emotions. Then, we consulted the definitions and their examples in the *Modern Chinese Dictionary* (7th Edition) [44], *Chinese Idioms Dictionary* (2nd Edition) [45], and *Chinese Idiom Advance Dictionary* [46] to make sure those embodied emotion words fell into the proper categories of emotion concepts. Furthermore, the embodied words and idioms used in the four great Chinese classics (i.e., *A Dream in Red Mansions*, *The Romance of Three Kingdoms*, *Outlaws of Marshes*, and *Journey to the West*) were collected, from which those associated with emotions were picked out and compared with their English translations based on the Chinese-English parallel corpus of Shaoxing University of Arts and Sciences, namely the Chinese-English parallel corpus of *A Dream of Red Mansions*, Chinese-English parallel corpus of *The Romance of Three Kingdoms*, Chinese-English parallel corpus of *Journey to the West*, and Chinese-English parallel corpus of *Outlaws of Marshes* (http://corpus.usx.edu.cn (accessed on 5 January 2022) [47], so that the embodied Chinese words and idioms could be compared with their English translations to hypothesize the rules underlying the conversion of emotion concepts across the two languages (e.g., whether or not the embodiments were maintained and which aspects of the embodiments were preoccupied by either language). The following are the findings from the collected data.

### 3.1. The Varied Embodiment of Fear in Chinese and English

In both Chinese and English, many words describing *fear* make reference to the physical reactions and expression of autonomic bodily responses (e.g., change in heart rate, temperature, sweating, and shaking of the body). Nevertheless, there appear to be more emotion words in general usage in Chinese compared with English that are coded with reference to interoceptive physiological changes and sensations attributed to specific internal organs (mainly the heart, gallbladder, and liver) to express *fear*.

Table 1 lists the Chinese words and idioms expressing *fear* with reference to physical reactions or reflexes controlled by the autonomic nervous system and visceral words such as ‘heart’ (xin), ‘gallbladder’ (dan), and ‘liver’ (gan) that are used for expressing *fear*.

Apparently, across multiple Chinese idioms, the expression of *fear* is embodied via agitation and trauma (such as shaking, trembling, dropping, tearing, splitting, and loss) of the internal organs such as the ‘heart’, ‘gallbladder’, and ‘liver’, often with reference to physical sensations attributed to these visceral organs (such as startled, panicked, broken, cold, frozen, chilly, weak, frightened, and timid).

#### Comparable Terms in English

The embodied words, idioms, and descriptions of *fear* and *fearful* states were collected from citations within *Roget’s 21st Century Thesaurus (the 7th edition)* [35] (p. 337) and from [35] (pp. 70–73) as well as two on-line sources: Collins thesaurus (https://www.collinsdictionary.com/us/dictionary/english-thesaurus/fear (accessed on 4 January 2022)) [36] and https://www.sketchengine.eu/skell (accessed on 4 January 2022) [37]. These terms are shown in Table 2:

As shown in Table 2, the above examples demonstrate that there are many embodied words in both the Chinese and English expressing *fear*, and only a few English verbal expressions for *fear* refer to the internal organs, yet many more words are associated with bodily parts and physiological activation controlled by the autonomic nervous system. Although in both languages such embodied emotion expressions for *fear* refer to internal bodily sensations and to physiological reactions controlled autonomically, words relating internal organs are used to a much greater extent and more systematically in Chinese when compared with English [4]. This increased granularity and transparency of using internal organs such as the ‘heart’, ‘gallbladder’, and ‘liver’ to label *fear* in Chinese may be attributed to the strong influence of traditional Chinese philosophy and medicine when compared with the paucity and piecemeal use of interoceptive words (e.g., heart, belly, stomach, and liver) in contemporary English [4].

### 3.2. The Embodied Conceptualization of Anger or Being Angry in English and Chinese

In Chinese, the majority of the words and idioms labeling *anger* are embodied, which means they are related to specific bodily sensations and actions, including facial expressions, skin complexion, physical reactions, or behaviors encompassing changes or agitation within visceral organs, notably the heart, liver, and lungs. In addition, there are numerous *anger* words that refer to natural phenomenon words such as *qi* (air), fire, wind, and thunder, as shown in Table 3.

Comparable terms in English can be found in Table 4.

As shown in Table 4, within the English-speaking North American culture, *anger* has been proposed to be metaphorically and metonymically conceptualized as output energy accumulated in the body as internal heat [18,20]. This may originate in a Western cultural understanding of physics, in which ‘*emotional effects are understood as physical effects. Anger is understood as a form of energy*’ [18] (p. 61). Thus, input energy accumulates within a body until it reaches a pressure point, at which the energy erupts as steam, externally radiating heat and agitation that may pose a danger to others.

However, within the same formulation, it is acknowledged that the ‘lexical approach’ toward mental structure (i.e., speculating about the mentalization of emotions via the words used in a particular language [18]) is likely to reflect more received ‘folk theories’ rather than the logic of scientific cognitive theories, particularly the updated modern affective neuroscience, even though Kövecses acknowledged those influential psychologists (e.g., [48,49,50]) by claiming that physiological reactions and bodily changes such as heat, internal pressure, redness of the face and neck area, and agitation are the essential components of angry emotion (and interfere with normal perception and reason [18]). Such theoretical logic is, however, somewhat obsolete and at odds with new evidence and emerging theories within affective neuroscience which highlight the fundamental, imperative role of interoception in emotional experiences.

The comparison of the embodied expressions of *anger* between Chinese and English (see Table 3; Table 4) demonstrates both similarities and differences in the two cultures. On the one hand, in each language, the facial expression, hair, teeth, eyes, and eyebrows, alongside physiological responses such as increased body temperature and redness of the face, are regarded as essential components of emotional experiences [18]. On the other hand, the Chinese and English languages differ in the following aspects: First, the way in which *anger* is typically conceptualized in English suggests a process that involves increasing temperatures within a *fluid* inside the body, leading to (implicitly through the evaporation) the build-up of pressure within the container (the body) and finally to the explosion of the container as a result of excessive pressure. *Anger* is construed more as the agitation of *qi* (in a gas or air state) in Chinese. This difference in the conceptualization of *anger* may be attributed to the distinct philosophical traditions of China and the West, in particular with regard to fundamental assumptions concerning the mind–body relationship [4].

More specifically, in traditional Chinese medicine and philosophy, everything in the universe is proposed to originate from the ever-changing and volatile primordial *qi*. In contrast, in early Western traditions, namely in the writings of Hippocrates, disease was associated with an imbalance or disturbance from the natural state of the body. In his *On the Nature of Man*, Hippocrates proposed the Theory of Four Humors, in which blood, phlegm, yellow bile, and black bile were the four elementary components of human bodies, and the imbalance or disproportion of the humors in the body may cause disease. Thus, a healthy state is conceived as the right balance in the intensity and quantity of the humors within the body. If one humor is insufficient or in excess, or if it is dispersed in the body and fails to mix with the others, disease will result [51]. Thus, in accordance with the theory of traditional Chinese medicine, the embodied words expressing *anger* that refer to the heart, liver, lungs, and other internal organs and to the flowing or circulation of *qi* between these internal organs are in compliance with the conception of mind–body relationships within Chinese philosophy [4,52], while references to bile and the spleen in English can be traced back to the historical origin of Western medicine [51,53].

### 3.3. The Embodied Conceptualization of Sadness or Grief in Chinese and English

A comparison of how *sadness* or *grief* is conceptualized in the Chinese and English languages showed the following: (1) There are far more words using tears and snot to express *grief* in Chinese than in English (merely with the more general word ‘weep’). (2) *Sadness* and *grief* terms in Chinese draw reference to trauma to and pain in the internal organs (notably the heart, lungs, liver, intestine, blood, and even all five viscera), while embodied *grief* in English is typically constrained to the heart. (3) The expression of *sadness* in Chinese idioms uses many *sadness*-related behaviors including wailing, lamenting, thumping one’s breast, and stamping one’s feet, as well as lamenting to heaven and knocking one’s head on the earth, but these are rarely mentioned in the English emotional language. (4) In Chinese, body parts associated with *sadness* or *grief* include the bone, bone marrow, skin, and eyes, while English lacks this specificity and granularity, using the general word ‘hurt’. (5) As for gustation, bitter and sour are the flavors for *sadness* in Chinese, while only bitter is used in English. (6) In terms of temperature sensing (thalposis), there are numerous words connected with coldness or chilliness in Chinese to express *sadness*. In addition, compound emotions are frequently produced by cold and other emotions, such as *qi can* (miserable = cold + wretched), *qi liang* (bleak = cold + cool or desolate), *qi qie* (plaintive = cold + sad), *qi ku* (miserable = cold + bitter), *qi wang* (desolate = cold + disappointed), qi shang (melancholy = cold + hurt), *qi mi* (gloomy = chilling + sorrowful), *qi chuang* (wretched = chilling + mournful), and *bei liang* (desolate = sad + chilling). In contrast, in English, these feelings are expressed with discrete abstract words such as bleak, desolate, sorrowful, mournful, miserable, and so on and so forth (see also [4]).

In short, comparatively, Chinese people tend to conceptualize *sadness* via physical perceptions (including exteroception and interoception), in addition to emotional behaviors, actions, and facial expressions, while in English, the lexicalization and conceptualization of *sadness* or *grief* is more impoverished, with a more limited range of words describing physical sensations, postures, and behavioral and facial expressions.

### 3.4. The Embodied Conceptualization of Joy or Happiness in Chinese and English

The comparison of the concept of *joy* or *happiness* between Chinese and English shows the following: (1) *Joy* is conceptualized as smiling, laughter, uncontrollable crazy behavior, celebration, excitement, and an energetic mental state in both languages. (2) Each language uses tactile sensations (e.g., itching) to describe *joy*. (3) Many Chinese idioms describing *joy* are underpinned by concepts of beaming, glowing, and radiance, such as *shen cai yi yi* (with shining and beaming spirit) (beaming), *guang cai zhao ren* (radiant with glamour and charm) (glamorous and charming), and *man mian chun feng* (the whole face in spring breeze) (overjoyed or beaming with joy). Likewise, in English, *joy* is conceptualized as glowing, radiance, and beaming of the face or body.

Nevertheless, there are variations in the conceptualization of *joy* between the two languages: (1) In Chinese, *joy* is mainly described with facial expressions (e.g., the stretching, lifting, and stirring of the eyebrows and eyes), postures and gestures (e.g., the shaking, stamping, and dancing of hands, feet, and the head) and bodily sensations including both somatosensation (e.g., itching) and interoception (e.g., *kai xin* (open heart) (joyful) or *xin hua nu fang* (flowers in the heart are in full bloom) (be elated or overjoyed)). Meanwhile, *joy* or *happiness* is less likely to be described with physical sensations in English, except for the itching and redness of the skin and the relaxation of the heart (e.g., heartening and lighthearted). (2) *Joy* is metaphorized as the abundance, fullness, freshness, smooth flowing, and stable state of *qi* inside the body in Chinese, while it is often conceptualized as lifting, flying, or floating of the body in the air in English.

In summation, the comparison of the four ‘basic’ emotions in Chinese and English indicates the following:Chinese uses more interoceptive words to describe emotions than in English. The Chinese emotion words with reference to the sensation and agitation of internal organs is not only abundant but systematic, likely due to the pervasive influence of traditional Chinese medicine and philosophy, whereas the adoption of interoceptive terms to describe emotions is not only far less common in English but also lacks granularity and systematicity (see also [4]).Under the influence of traditional Chinese medicine and philosophy, many Chinese emotion words are associated with *qi*, while in English, emotions such as *anger*, under the impact of the ancient theory of humors, are viewed as the changing energy of *fluids* inside the body and the increase in their temperature, vaporization, expansion, and explosion.Generally, ‘coldness’ or ‘chill’ is metaphorically projected to the concept of *sadness* in Chinese. This cold sensation, when combined with other feelings, generates more complex emotions such as bleakness, desolation, sorrowfulness, mournfulness, misery, and melancholy. In contrast, the sense of being chilled is more directly connected with *fear* in English.Incidentally, as pointed out elsewhere, there are far more emotion words and phrases using bodily sensory-motor systems such as facial expressions, bodily movements, and internal and external sensations in Chinese than in English, in which emotions are more likely to be conceptualized with nuanced abstract concepts [4].

In short, the interoception-centered embodiment of emotion concepts and their lexicalization in Chinese encapsulate the holistic body–mind–emotion relationship of traditional Chinese medicine and philosophy [4,52]. In contrast, a more dichotomous model of body–mind interaction underpins the assumptions of Western philosophy regarding the role of the body in emotion.

Therefore, what might be the impact of this divergence in embodied emotion concepts in Chinese and English on the everyday perception and experience of emotions? Moreover, do such linguistically diverse conceptual systems for emotions correspondingly shape or nurture distinct cultural values expressed by groups of Chinese and English language users as suggested by the Sapir–Whorf hypothesis and the theory of constructed emotion (e.g., [16,33,54])? We propose a tentative hypothesis that the prominence given in Chinese to interoception (i.e., the cerebral sensory representation of inner bodily processes and the feeling states that are generated through this afferent body-to-brain route in the conceptualization of emotions) places bodily sensations underlying emotions in the foreground for the mind to receive, process and adapt to. In contrast, the prominence given in English to physical actions controlled by the autonomic nervous system and to reactive behaviors transmitted along the efferent brain-to-body pathway implies that bodily reactions are the principal expression of the embodiment of emotions and are subject to overarching control by the brain (and mind). In this latter context, across Western culture, the brain is viewed as the ‘master’ or ‘commander-in-chief’ that plays a steering and directing role in emotion, while the body is reactive and subservient to the brain’s wishes within the affective brain–body dynamics.

Arguably, the idiosyncratic embodiment of emotion concepts in the two geoculturally remote languages (i.e., Chinese and English) may be attributed to their distinctive conceptions of the body out of their distinct cultural or civilizational origins. In other words, the divergence in how the body is conceived can primarily explain the structural and systematic variation in the embodied conceptualization of emotions between English and Chinese (e.g., [16]). Next, we will delve into the impact of conceptions of the body on the embodiment of emotion concepts across local cultures and the characterization of cultural values and personalities.

## 4. Conceptions of the Body and the Culture-Loaded Embodiment of Emotion Concepts

The conception of the body is argued to be a construction of specific cultures rather than being universal across cultures and languages [15]. The body is arguably construed as ‘muscles’ across Western cultures, while it is conceived as interconnected meridians (i.e., tracts and points of acupuncture) in traditional Chinese medicine. Accordingly, cultural or linguistic idiosyncratic notions of the body may, to some extent, shape distinctive emotion concepts and furthermore characterize the values and personalities of those cultures [15].

Embodied cognitive theories presume that the body is an essential constituent of mental activities; that is to say, our thoughts and minds are derived from and constrained by our physical attributes and capabilities (‘No body, no mind’) [55]. As a consequence, embodiment shapes the profile of knowledge and the way in which we understand meanings [56,57,58]. However, the specific nature of how the body *per se* is represented varies across different cultures [15,56,59,60,61], rather than being universal, as was assumed by some researchers (e.g., [19]). For instance, traditional Chinese philosophy is holistic in presuming that the body and *xin* (heart or mind) are integral or inseparable. Contemporary Western cultures typically take a more dualistic view and assume the dichotomy of the body and mind. Thus, concepts such as *human*, *self*, and *agent* are construed differently and show systematic or structural differences between Chinese and Western cultures [62,63,64,65]. Furthermore, such differences may have archaic origins in concepts of the body from ancient Chinese and in ancient Greek medicine [15]. Chinese medical teachings emphasize the systematic, interrelated, and interactive nature of the body, distributed with acupuncture tracts and points and with little attention to muscular detail. In the Western tradition, muscularity has been more of a preoccupation, yet the meridian tracts and points entirely escaped this particular anatomical vision of reality. 

Hence, the Chinese view of the body is argued to be less anatomical (and perhaps more functional) than the ancient Greek conceptions of the body [15]. The anatomical notion of muscles, distinct from flesh, tendons, and sinews, developed uniquely in ancient Greece in contrast to other old medical traditions (i.e., Egyptian and Avurvedic, in addition to Traditional Chinese Medicine) that flourished for thousands of years without that same interest in anatomical inquiry or an emphasis on muscles [15].

Galen observed two categories of bodily activity: involuntary processes and voluntary actions. The former are the internal processes such as digestion and pulsation, over which we exert minimal direct control or influence, despite our intentions. Voluntary processes, in contrast, encompass activities such as walking and talking, which are subject to our desire and intentions. For example, we can choose to do something such as walking or running, during which we can change the speed of our movement and adjust the intonation of our speech because our muscles are organs of voluntary action. Muscles, in short, enable us to act as a genuine agent, supporting self-awareness and volition. Thus, rather than passive containers of visceral organs enacting involuntary processes such as digestion and pulsation, the primary identification of humans as muscular creatures implies that they are agents who execute intentional actions on the world [15].

This distinction runs deeper when considering explanations of the causes of diseases. In traditional Chinese medicine, dysfunction of the internal organs is regarded as the first and foremost cause of diseases, expressed and accessed via seven external apertures on the human head and meridians on the body (skeletomusculature overlooked or unnoticed) [15,52]. Instead, ancient Greek explanations of the causes of diseases focus on the external muscular body, which epitomized the beauty of the human form, while the viscera were neglected, viewed to be as unclean as dead corpses. It was therefore unnecessary to seek the causes of diseases from the inner body [53].

Thus, even though human beings broadly share the same physiology, structure, and functions of the body, cultural differences in how the body is perceived and conceived differ radically across cultures. In ancient China, the body, emotions, mind and spirit are understood as an interdependent and interactive integral whole that can be unified and correspond with the reality and the universe. Yet, in the Western cultures, through an emphasis on anatomy dating back to ancient Greek medicine, the body is viewed as an object that can be observed and examined objectively and treated as independent from the mind.

In summation, the historical preoccupation of Western thought with muscularity underpins the dichotomy between the body and mind. By extension, physiology and behavior can presumably be subsumed into two contrasting processes: one happens naturally or incidentally, and the other is ‘controlled by the soul’ [53].

The very concept of ‘muscle’ as the essential instrument for voluntary action is omitted in traditional Chinese culture, especially when considering emotions. However, the physiological reactions evoked by the autonomic nervous system and associated voluntary physical actions feature strongly in the Western conceptualization of embodied emotions. Consequently, Chinese discourse or narrations generally underappreciate or unvalue many Western cultural themes, such as free will and self-awareness, being linked systematically and conceptually to muscularity. There has perhaps been some consolidation of this stance, as in the ancient Pre-Qin period of Chinese history (before 221 BC), a tall and well-built male physique was favored as encouraging a strong military spirit. However, such aesthetic criteria had dramatically shifted in the opposite direction over the course of the Wei and Jin North and South Dynasties (220–589 AD) until the fall of the Qing Dynasty (in 1912), when rather than advocating physical fitness, the image of a frail-looking or pale-complexioned scholar was regarded as the standard (even ideal) model for the male physique. There is probably some debt owed to the mapping of *Yin* and *Yang* cosmology and the Chinese feudalistic hierarchical system. Here, emperors represent *Yang*, while ministers represent *Yin*. Therefore, for emperors, ministers are expected to have slighter physiques and to ensure loyalty and prolong the stability of their rule [66,67].

Conceptions of the body can fundamentally mold peoples’ views on emotions. The preoccupation of muscularity in the Western notion of the body may consequently explain the frequent use of reactive or proactive verbs such as ‘combat’, ‘fight’, ‘overcome’, ‘prevent’, ‘conquer’, or ‘assuage’ together with negative emotion concepts such as *fear*, *anger,* and *sadness* in English, owing to the fact that the metaphor NEGATIVE EMOTIONS ARE ENEMIES underpins the conceptualization of negative emotions in English. Taking *fear* as an example, the notion that it should be battled is apparent:Overcome fear: Cognitive behavioral therapy helps people *overcome fear;*(https://skell.sketchengine.co.uk/run.cgi/wordsketch_concordance?headword=fear-n;gramrel=verbs%20with%20%w%20as%20object;coll=overcome-v) (accessed on 26 June 2022) [37];Prevent panic: Such symbols can play an important role in reassuring the public and *preventing panic*;(https://skell.sketchengine.co.uk/run.cgi/wordsketch_concordance?headword=panic-n;gramrel=verbs%20with%20%w%20as%20object;coll=prevent-v) (accessed on 24 June 2022) [37];Assuage fear: A letter from Superintendent Julian Field *assuaged the fears* of most members;(https://skell.sketchengine.co.uk/run.cgi/wordsketch_concordance?headword = fear-n;gramrel = verbs%20with%20%w%20as%20object;coll = assuage-v) (accessed on 24 June 2022) [37].

This reactive and proactive framing of emotion is largely absent in Chinese where, arguably, the neglect of muscle with an aesthetic bias toward emaciation, together with a preoccupation with internal organs in traditional Chinese medicine, has arguably led to the pervasive use of interoceptive words to describe emotions in the Chinese language.

Incidentally, the abundance in Chinese emotion words with *qi* can be attributed to the influence of the theory of *qi* in traditional Chinese medicine, such as *xi qi yang yang* (voluminous joyful *qi*) (beaming with joy), *nu qi chong tian* (towering *qi* rushes to the sky) (one’s wrath filled the sky or to be in a towering rage), *chui tou sang qi* (bow one’s head and crestfallen *qi*) (down in the dumps, crestfallen, singing the blues, or down in the mouth), *yang mei tu qi* (raise eyebrows and give vent to *qi*) (feel proud and elated), *xin ping qi he* (heart in peace and *qi* in harmony) (even-tempered and good humored or a heart at ease), *sheng qi ling ren* (domineering *qi* and bullying others) (domineering or airs and graces or overbearing), *qi ding shen xian* (stable *qi* and leisurely spirit) (calm and peaceful), and so on and so forth. The bodily *qi* is generated through the coordinating movement and circulation between internal organs such as the spleen, kidneys, lungs, and other organs [68]. Specifically, when the *qi* flows smoothly through the body and the rise and fall and in and out of *qi* within the body are in a balanced and harmonious state, this is called ‘harmonious *qi*’ in traditional Chinese medicine. In contrast, when the circulation of *qi* is slowed, blocked, or in the wrong direction, it is called ‘disharmonious *qi*’, mainly manifested as ‘stagnancy of *qi* activity’ (the movement of *qi* being blocked), ‘blockage of *qi*’ (*qi* is blocked), ‘circulation of *qi* in the wrong direction’ (the increase in *qi* is too high, or the fall is too low), ‘*qi* collapse’ (the increase in *qi* is too high, or the fall is too low), ‘*qi* exhaustion’ (an excessive outflow of *qi* that cannot be kept inside), and ‘*qi* closure’ (*qi* cannot reach outside and is blocked inside) [68].

Thus, from the perspective of traditional Chinese medicine, emotions are an outcome of the movement and circulation of the essence and *qi* between internal organs in response to external stimulation. The essence of the five internal organs can correspondingly produce five kinds of emotional activities. For instance, the heart generates *joy*, the liver *anger*, the spleen *sorrow*, and the kidneys *fear* [52,68]. Conversely, when the external stimulation is so intense that it causes excessive or persistent emotional fluctuation, this may lead to an imbalance of the *yin* and *yang qi* within the internal organs and the dysfunction of *qi* blood circulation. For instance, excessive joyfulness impairs the heart *qi*, excessive anger impairs the liver *qi*, anxiety impairs the spleen *qi*, and fear impairs the kidney *qi* [52,68].

In short, within traditional Chinese medicine, the internal organs, emotions, and conscious mind are viewed as interactive, interdependent, and integrated facets of the same fundamental entity. Essential *qi* is stored in the five key internal organs (i.e., the heart, liver, spleen, lungs and kidneys), which may engender the corresponding emotions of *joy*, *anger*, *pensiveness*, *sadness,* and *fear* in response to excessive stimulation from the external environment. Immoderate expression of emotions may impair the balanced circulation of *qi*, which may in turn cause visceral dysfunction. Conversely, within this framework, the inharmonious circulation of *qi* between the internal organs is proposed to give rise to emotional and mental disorders. Interestingly, these embodied theories of emotions parallel the emerging findings from modern interoceptive neuroscience and affective science [7,8,9,69,70,71].

## 5. Conclusions

Traditional cognitive science applied to emotion has typically overlooked the role of interoception as the substrate for subjective awareness of emotion. Even models of embodied emotion have failed to differentiate the afferent and efferent pathways of signaling between the body and the brain. Consequently, dominant cognitive linguistic theories also do not make such distinctions and cannot account for a variety of concepts across languages and cultures applied to the embodiment of emotions. Advances in affective psychology and interoceptive neuroscience, however, indicate that affective feelings emerge from interoceptive signals that are integrated and represented within specific interconnected brain regions (notably the insula cortex) and are translated into subjective awareness and heuristic emotional concepts with varying degrees of bodily transparency and cognitive granularity, being shaped by sociocultural nuances [4]. Although in both the Chinese and English languages, emotions are described with reference to bodily organs and physiological functions, we highlight how these languages differ with respect to the distinct emphasis on afferent versus efferent signaling in brain–body interaction and argue that Chinese and English diverge in how emotion concepts are embodied. In Chinese, emotion concepts are more strongly linked to feelings arising from afferent representations of interoceptive states. However, in English, the overt physical and behavioral expressions of emotion, including physiological reactions evoked by efferent autonomic pathways, are foregrounded. By extension, divergent conceptual systems of emotions within languages may foster distinct cultural attitudes and narratives regarding emotions. Chinese speakers may be biased toward being receptive, reflective, and adaptive in how emotions impact lives and decisions, whereas English speakers, as well as others who share the same Western philosophical tradition, may be biased toward a more reactive and proactive stance in their affective and emotional behaviors.

## Figures and Tables

**Table 1 brainsci-12-00911-t001:** Chinese embodied words and idioms labeling bodily states of *fear*.

Chinese Embodied Words and Idioms Labeling Fearful Bodily States	
Bodily States Controlled by the Autonomic Nervous System	
Fear as changes in complexion	*mian wu ren se* (face without a human’s color): as pale as death *da jing shi se* (losing color out of immense shock): turn pale with fright *jing kong shi se* (losing color out of shock and fright): pale with fear *lian se fa qing* (face blue in color): be overly scared *lian se sha bai* (complexion is deadly pale): turn pale with fright
Fear expressed in eyes and mouth	*mu deng kou dai* (eyes staring and mouth stupefied): stunned *mu deng kou jiang* (eyes staring and mouth frozen): dumbstruck *cheng mu jie she* (eyebrows rising and tongue tied): stare dumbfounded *cheng mu er shi* (raising eyebrows to see): stare at with wide eyes *zui chun fa bai* (white-lipped): frightened with lips turning pale or colorless
Fear as changes in hair and bone	*mao gu song ran* (with one’s hair and bones horrified or with one’s hair standing on end): shivers or being bloodcurdling *han mao dao shu* (with hair erected): very frightened *gu han mao shu* (bone chills and hair stands up): make one’s blood run cold *ji liang gu mao liang qi* (send chilly *qi* up somebody’s spine): absolutely terrified
Fear as changes in skin	*qi ji pi ge da* (with chicken bumps): goose bumps
Fear in excretion of body fluids (sweat, urine, etc.)	*xia de pi gun niao liu* (so frightened that one’s fart rolls and urine flows): scare the shit out of someone, be frightened out of one’s wits, piss one’s pants (in terror), or wet one’s pants in terror *zhi mao leng han* (cold sweat runs out): sweat bursts out in fear *nie yi ba han* (pinch a handful of sweat): break into a sweat with fright (fear) or be breathless with anxiety or tension *yi shen leng han* (be wet with cold sweat): be wet with cold sweat, be soaked in cold and clammy perspiration, be in a cold (icy) sweat, break out in a cold sweat, a cold sweat breaks out all over one’s body, or one’s body is covered with chilly sweat
Fear as body quivering	*xia de hun shen fa dou* (tremble from head to foot with fear): be all of a tremble, tremble with every inch of one’s body, trembling all over, or trembling in every limb out of fear
Bodily Sensations Governed by the Interoceptive System	
Xin (heart)	*xin you yu ji* (heart still fluttering): have a lingering fear, or still being in a state of shock *chu mu jing xin* (touch the eyes and shock the heart): strike the eyes and rouse the mind, shocking, or startling
Dan (gallbladder)	*hun fei dan sang* (spirit flies and gall is lost): strike terror in one’s heart
Dan and Xin (gallbladder and heart)	*dan po xin jing* (gall broken and heart startled): startled *xin dan ju lie* (the heart and gall are broken into pieces): be frightened out of one’s wits, be heart-broken and terror-stricken, lost in great astonishment, be so frightened that one’s heart and galls burst, or terror-struck *xin han dan luo* (heart is frozen and gall falls to the ground): be extremely terrified or terror-stricken
Gan and Dan (liver and gallbladder)	*gan dan ju lie* (one’s liver and gall both seemed torn from within): extremely frightened, heart-broken, terror-stricken, or overwhelmed by grief or terror

**Table 2 brainsci-12-00911-t002:** English embodied words describing *fear* or fearful bodily states.

Bodily States Controlled by Autonomic Nervous System	
Fear as changes in complexion	e.g., She turned pale. You are white as a sheet.
Fear as inability to move	Paralyzed, stunned, weak-kneed e.g., I was rooted to the spot. He was so terrified he could not move.
Fear as inability to breathe	e.g., She was breathless or gasped in fear.
Fear as inability to speak	Dumbstruck, gape, tongue-tied, tongue stands still e.g., I was speechless or dumb with fear.
Fear as dysfunction in nerves	Nerveless, nervous, nervy, nerve-wracking, spineless
Fear as shrinking sensations in skin	Goosebumps, creeps e.g., That man gives me the creeps. A shriek in the dark gave me goosebumps.
Fear as hair straightens out	e.g., The story of the murder made my hair stand on end. That was a hair-raising experience.
Fear as drop in body temperature	Cold sweat, cold feet, blood-curdling, bone-chilling e.g., Just the face of the monster was enough to make my blood run cold. I heard a blood-curdling scream. A cold sweat of fear broke out.
Fear as body quivering	Agitation, heebie-jeebies, jitters, jumpy, quivery, shaky, trembling, tremor, tremulous, trepidation
Fear as (involuntary) release of bowels or bladder	e.g., I was scared shitless when I saw the man with the knife coming toward me. I was almost wetting myself with fear.
Fear as dryness in the mouth	e.g., My mouth was dry when it was my turn. He was scared spitless.
**Visceral Sensations Governed by the Interoceptive System**	
Heart	Chickenhearted, fainthearted, making someone’s heart leap or one’s heart gallop, heart in the boots, heart stood still, heart pounding, strike fear into the hearts of, terror into somebody’s’s heart, heart in one’s month e.g., His heart pounded with fear. My heart began to race when I saw the animal. His heart stopped or missed a beat when the animal jumped in front of him
Stomach	Butterflies in the stomach, collywobbles e.g., He got butterflies in his stomach. A cold fear gripped him in the stomach. I always get the collywobbles before an interview. Her husband went climbing mountains last weekend. It gave her the collywobbles to even think about it
Belly	Yellow belly e.g., My friend has a female yellow-bellied slider. This was no time for being some pasty yellow-bellied mama’s boy.
Liver	Lily-livered e.g., She approaches songs and arrangements with a sense of adventure that makes almost everybody else sound lily-livered. We have lily-livered textbook publishers whose toned-down presentations pander to the worst of our society.

**Table 3 brainsci-12-00911-t003:** Chinese embodied words and idioms labeling bodily states of anger.

Chinese Embodied Words and Idioms Labeling Bodily States of Anger	
Bodily States Controlled by the Autonomic Nervous System	
Anger in facial expressions, bodily reactions and/or behaviors	*chi mian* (red faced): catch fire *zhe mian* (with a reddish brown face): very angry *yun rong* (an angry look): in a sulk *nu se* (an angry look): wear an angry look or look black *li se* (harsh countenance): stern *yao ya* (grit one’s teeth): grind one’s teeth in anger *qi de lian sha bai* (face is deadly pale with angry qi): get red with anger *lian hong bo zi cu* (with one’s face red and neck swollen): one’s face turns crimson (red) with anger, being red to the tip of one’s ears, blue in the face, or flush with agitation (fury), get red in the face from anger or excitement or red in the face and fuming, or turn red in the gills *zha mao* (with hair stands up): blow up *chen mu e wan* (stare angrily and wring one’s wrist): angry and courageous *heng mei leng yan* (flattened eyebrows and cold face): frown and look coldly *ji zhi nu mu* (point one’s fingers at somebody and stare at him with angry eyes): point and look at somebody furiously *chen mu qie chi* (staring the eyes and gritting the teeth): staring and gritting with anger *fa zhi zhi lie* (with hair standing up and eye sockets tearing): boil with anger *liu mei dao shu* (willow leaf-shaped eyebrows risen): raise one’s eyebrows in anger
Bodily Sensations Governed by the Interoceptive System	
Anger as the feeling of physical changes in the visceral organs	*nu cong xin tou qi, e xiang dan bian sheng* (anger springs from the heart, and evil grows to the gall): be furious and nurse thoughts of revenge *da dong gan huo* (violently stirred the liver fire): fly into a rage or hit the roof *ji huo gong xin* (acute fire attacks the heart): burn with anger *fei qi zha le* (the lungs exploded with *qi*): burst with rage
Anger as the agitation of *qi* inside the body	*qi de tiao jiao* (with so much *qi* that one stamps): stamp one’s feet with anger *fa pi qi* (*qi* in the spleen exploded): lose one’s temper *sheng qi* (generating *qi*): anger or getting angry *sheng men qi* (generating silent *qi*): be in a sulk *nu qi* (angry *qi*): anger, rage, or fury *ou qi* (be repressed with *qi*): sulk or repressed grievances *nu qi chong chong* (angry *qi* rushes out): huff and puff, be in a fit of spleen, in a great rage, or in a huff, or seethe with anger *qi fen tian ying* (the breast is filled with angry ***qi***): be filled with indignation

**Table 4 brainsci-12-00911-t004:** English embodied words and idioms describing bodily states or sensations of anger.

English Embodied Words and Idioms Describing Bodily States or Sensations of Anger	
Bodily States Controlled by the Autonomic Nervous System	
Anger as the output energy accumulated in the body as internal fluid heat and evaporation pressure	*Heated*, *hot*, *slow burn*, *incensed*, *stew*, *blow up*, *fuming*, *inflame* e.g., Do not get *hot under the collar*. Billy’s a *hothead.* They were having a *heated* argument. When the cop gave her a ticket, she got all *hot and bothered*. Do not get a *hernia*! When I found out, I almost *burst a blood vessel*. He almost had a *hemorrhage*.
Anger as bodily injury or unpleasant bodily sensations	*Cat fit*, *fit*, *rankling*, *inflamed*, *convulsed*, *exacerbated*, *nettled*, *chafed*, *sore or soreness*, *bitter*
Anger as redness in face and neck area	She was *scarlet* with rage. He got *red* with anger. He was *flushed* with anger.
Anger as agitation	She was *shaking* with anger. I was *hopping* mad. He was *quivering* with rage. He is all *worked up*. She is all *wrought up*.
Anger as interference with accurate perception	She was *blind* with rage. I was beginning to *see red*. I was so mad I *could not see straight.*
Anger as breath or noise made by breath	huff, huffy, hissy
Bodily Sensations of Anger Governed by the Interoceptive System	
Anger as physiological sensation and changes in the visceral organs	*Choler*, *gall*, *ill humor*, *choleric*, *galled*, *splenetic*

## Data Availability

Data sources are described in full within the paper. The corresponding author can be contacted for further details.

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
