# Peer review of "Divergent Conceptualization of Embodied Emotions in the English and Chinese Languages"

_brainsci, 2022, doi:10.3390/brainsci12070911_

Round 1
Reviewer 1 Report
The article introduces an interesting topic. However, some modifications should be introduced to make sure that the arguments provided are solid and all the information given is clear.
Introduction & State of the Art
- The claim made about the three broad categories of language (p. 2) should be explained in more detailed in order for the reader to understand how this classification was made. Otherwise, we do not know which the basis was.
Methodology
- It does not precise how the words or idioms listed for both Chinese and English were detected using thesauri.
- Even of the thesaurus used to identify the words and idioms in English is mentioned in the text, the source of Chinese words and idioms is not stated.
- The data is not offered in a systematic way, since in English words and idioms appear separately, but this distinction is no offer for Chinese language. In this sense, if there is something about the structure and pattern of Chinese words and idioms that the reader needs to know, an explanation should be given.
- Chinese expressions for each emotion (fear, joy, etc.) are always classified in different subgroups, but the English data is not always offered in the same way. For instance, in the case of fear.
To sum up, a clear and more detailed methodology should be offered and applied to extract, categorize and analyse the data, and it should be systematic in both languages in order to contrast the results obtained.
Analysis
- The article seems to lead into the consideration that data will be quantified, but it is not the case. If it were done, the "pattern" would be more supported, since now the conclusions extracted from the analysis are not clearly justified.
Style
- Some sentences are too long and are not well connected between them.
- There are some minor mistakes (missing and unnecessary spaces)
- The numbers preceding some parts of the manuscript are not correct.
- Some of the references should be revise because book titles need to be in italics, for example.
Author Response
Cover Letter for revised Manuscript ID: brainsci-1738668:
Divergent conceptualization of embodied emotions in English and Chinese languages
Dear Editors for
Review #1
The article introduces an interesting topic. However, some modifications should be introduced to make sure that the arguments provided are solid and all the information given is clear.
Introduction & State of the Art
Point 1:
The claim made about the three broad categories of language (p. 2) should be explained in more detailed in order for the reader to understand how this classification was made. Otherwise, we do not know which the basis was.
Response 1:
We appreciate with the insightful comments and helpful suggestions for revision. We have explained in more detail, with references, the three general categories for culture-specific conceptualization of emotions and mind in relation to global geography on page 2, which reads as follows:
Some researchers generalize three broad conceptualization of the mind and emotion categories globally as: (1) Abdominocentrism (i.e., the mind, including feeling, thinking and knowing, is conceptualized as being located in or around the abdomen region, e.g. in the belly, in the liver or in the kidney). Southern Asia, Polynesia and other disparate cultures, including Basque culture are the prototypical abdominocentric cultures. (2) Cardiocentrism (i.e., the mind and emotions are located in the heart region). Cardiocentrism is the traditional view of China, Korea, and Japan, with their shared similar philosophical and medical cultural models of holistic, heart-centering conceptualizations.)(3) Cerebrocentrism (i.e., the mind is believed to be seated in the head or the brain region). This conception of the mind is mainly held by Greek-based West Asian, European, and North-African cultures, with the prototypical examples being the major Indo-European languages. [39, 48]
Point 2:
Methodology
- It does not precise how the words or idioms listed for both Chinese and English were detected using thesauri.
- Even of the thesaurus used to identify the words and idioms in English is mentioned in the text, the source of Chinese words and idioms is not stated.
- The data is not offered in a systematic way, since in English words and idioms appear separately, but this distinction is no offer for Chinese language. In this sense, if there is something about the structure and pattern of Chinese words and idioms that the reader needs to know, an explanation should be given.
- Chinese expressions for each emotion (fear, joy, etc.) are always classified in different subgroups, but the English data is not always offered in the same way. For instance, in the case of fear.
To sum up, a clear and more detailed methodology should be offered and applied to extract, categorize and analyse the data, and it should be systematic in both languages in order to contrast the results obtained.
Response 2:
The reviewers comments on the methodology are pertinent and to the point. We have added the source of the data on page 4, detailing our approach. The revised text reads as follows:
To explain further this point, the English words labeling the four so-called ‘basic’ emotions, namely, fear, anger, sadness and joy were collected from Roget’s 21st Century Thesaurus (the 7th edition) and on-line dictionary such as [26]. The Chinese words and idioms expressing the four ‘basic’ emotions and the bodily words that are associated with emotions were collected and selected from the Modern Chinese Dictionary [13] (7th Edition), Chinese Idioms Dictionary (2nd Edition)[64], Chinese Idiom Advance Dictionary [36], and from selected on-line dictionaries, specifically http://zhidao.baidu.com, http://xh.5156edu.com, http://chengyu.t086.com and http://www.hydcd.com. Identified emotion words and idioms were then compared between Chinese and English.
Moreover, to further clarify this approach, we have reorganized the data into 4 Tables as shown below:
Table 1. Chinese embodied words and idioms labeling bodily states of fear (pp. 5-6)
|
Chinese embodied words and idioms labeling fearful bodily states |
|
|
Bodily states controlled by autonomic nervous system |
|
|
Fear as changes in complexion |
MIAN WU REN SE (face without human’s color): as pale as death; DA JING SHI SE (losing color out of immense shock): turn pale with fright; JING KONG SHI SE (losing color out of shock and fright): pale with fear; LIAN SE FA QING (face in blue color): be over-scared; LIAN SE SA BAI (complexion is deadly pale): turn pale with fright... |
|
Fear expressed in eyes and mouth |
MU DENG KOU DAI (eyes staring and mouth stupefied): stunned; MU DENG KOU JIANG (eyes staring and mouth frozen): dumbstruck; CHENG MU JIE SHE (eyebrows rising and tongue tied): stare dumbfounded; CHENG MU ER SHI (raising eyebrows to see): stare at with wide eyes; ZUI CHUN FA BAI (white lipped): frightened with lips turn pale or colorless... |
|
Fear as changes in hair and bone |
MAO GU SONG RAN (with one's hair and bones horrified/with one's hair standing on end): shivers or be bloodcurdling; HAN MAO DAO SHU (with hair erected): very frightened; GU HAN MAO SHU (bone chills and hair stands up): make one’s blood run cold; JI LIANG GU MAO LIANG QI (send chilly qi up somebody's spine): absolutely terrified... |
|
Fear as changes in skin |
QI JI PI GE DA (with chicken bumps): goose bumps... |
|
Fear in excretion of body fluids (e.g., sweat, urine, etc.) |
XIA DE PI GUN NIAO LIU (so frightened that one’s fart roll and urine flow): scare the shit out of someone/be frightened out of one's wits/piss one's pants (in terror)/wet one's pants in terror; ZHI MAO LENG HAN (cold sweat runs out): sweat bursts out in fear; NIE YI BA HAN (pinch a handful of sweat): break into a sweat with fright [fear]’/be breathless with anxiety or tension; YI SHEN LENG HAN (be wet with cold sweat): be wet with cold sweat/be soaked in cold and clammy perspiration/be in a cold [icy] sweat/break out in cold sweat/cold sweat breaks out all over one's body/one's body is covered with chilly sweat... |
|
Fear as body quivering |
XIA DE HUN SHEN FA DOU (tremble from head to foot with fear): be all of a tremble/tremble with every inch of one's body/trembling all over/ trembling in every limb out of fear... |
|
Bodily sensations governed by interoceptive system |
|
|
Xin (heart) |
XIN YOU YU JI (hear still fluttering): have a lingering fear/be still in a state of shock; CHU MU JING XIN (touch the eyes and shock the heart): strike the eyes and rouse the mind/shocking/startling... |
|
Dan (gallbladder) |
HUN FEI DAN SANG (spirit flies and gall is lost): strike terror in one's heart... |
|
Dan and Xin (gallbladder and heart) |
DAN PO XIN JING (gall broken and heart startled): startled; XIN DAN JU LIE (the heart and gall are broken into pieces): be frightened out of one's wits/be heart-broken and terror-stricken/lost in great astonishment; be so frightened that one's heart and galls burst; terror-struck; XIN HAN DAN LUO (heart is frozen and gall falls to the ground) : be extremely terrified/terror-stricken... |
|
Gan and Dan (liver and gallbladder) |
GAN DAN JU LIE (One's liver and gall both seemed torn from within): extremely frightened/heart-broken or terror-stricken/overwhelmed by grief or terror... |
Comparable terms in English
The embodied words, idioms, and descriptions of fear and fearful states were collected from citations within Roget’s 21st Century Thesaurus [24] (p.337) and from [26] (pp.70-73)and two on-line sources: Collins thesaurus https://www.collinsdictionary.com/us/dictionary/english-thesaurus/fear, and, https://www.sketchengine.eu/skell. These terms are shown in Table 2:
Table 2. English embodied words describing fear /fearful bodily states (pp. 6-7)
|
English Embodied words and idioms describing fearful bodily states |
|
|
Bodily states controlled by autonomic nervous system |
|
|
Fear as changes in complexion |
E.g., She turned pale./You are white as a sheet. |
|
Fear as inability to move |
paralyzed, stunned, weak-kneed E.g., I was rooted to the spot./He was so terrified he couldn't move. |
|
Fear as inability to breathe |
E.g., She was breathless /gasped with fear. |
|
Fear as inability to speak |
dumbstruck, gape, tongue-tied, tongue stand still E.g., I was speechless/dumb with fear. |
|
Fear as dysfunction in nerves |
nerveless, nervous, nervy, nerve-wracking, spineless... |
|
Fear as shrinking sensations in skin |
goosebumpy, creeps E.g., That man gives me the creeps. /A shriek in the dark gave me goosebumps. |
|
Fear as hair straightens out |
E.g., The story of the murder made my hair stand on end./That was a hair-raising experience. |
|
Fear as drop in body temperature |
cold sweat, cold feet, blood curdling, bone chilling... E.g., Just the face of the monster was enough to make my blood run cold. /I heard a blood-curdling scream./The cold sweat of fear broke out. |
|
Fear as body quivering |
agitation, heebie-jeebies, jitters, jumpy, quivery, shaky, trembling, tremor, tremulous, trepidation... |
|
Fear as (involuntary) release of bowels or bladder |
E.g., I was scared shitless when I saw the man with the knife coming towards me./I was almost wetting myself with fear. |
|
Fear as dryness in the mouth |
E.g., My mouth was dry when it was my turn./He was scared spitless. |
|
Visceral sensations governed by interoceptive system |
|
|
Heart |
chickenhearted, fainthearted, making someone’s heart leap, one’s heart gallop, heart in the boots, heart stood still, heart pounding, strike fear into the hearts of , terror into somebody’s’s heart, heart in one’s month E.g., His heart pounded with fear./My heart began to race when I saw the animal./His heart stopped/miss a beat when the animal jumped in front of him... |
|
Stomach |
butterflies in the stomach, collywobbles E.g., He got butterflies in the stomach./A cold fear gripped him in the stomach. /I always get the collywobbles before an interview. / Her husband went climbing mountains last weekend. It gave her the collywobbles even to think about it... |
|
Belly |
yellow belly E.g., My friend has a female yellow-bellied slider. /This was no time for being some pasty yellow-bellied mama's boy. |
|
Liver |
lily-livered E.g., She approaches songs and arrangements with a sense of adventure that makes almost everybody else sound lily-livered. /And we have lily-livered textbook publishers whose toned-down presentations pander to the worst of our society. |
In Chinese, the majority of the words and idioms labeling anger are embodied, which means they are related to specific bodily sensations and actions, including facial expressions, skin complexion, physical reactions and/or behaviors encompassing changes or agitation within visceral organs notably heart, liver, and lungs. In addition, there are numerous anger words that refer to natural phenomenon words such as qi (air), fire, wind and thunder as shown in Table 3.
Table 3. Chinese embodied words and idioms labeling bodily states of anger
|
Chinese embodied words and idioms labeling bodily states of anger |
|
|
Bodily states controlled by autonomic nervous system |
|
|
Anger in facial expressions, bodily reactions and/or behaviors |
CHI MIAN (red faced) : catch fire, ZHE MIAN (with reddish brown face) : very angry, YUN RONG (an angry look): in a sulk, NU SE (an angry look) : wear an angry look/look black, LI SE (harsh countenance): stern, YAO YA (grit one’s teeth): grind one's teeth in anger, QI DE LIAN SHA BAI (face is deadly pale with angry qi) : get red with anger, LIAN HONG BO ZI CU (with one's face red and neck swollen) : one's face turns crimson [red] with anger/be red to the tip of one's ears/blue in the face/flush with agitation [fury]/get red in the face from anger or excitement/red in the face and fuming/turn red in the gills), ZHA MAO (with hair stands up): blow up, CHEN MU E WAN (stare angrily and wring one’s wrist): angry and courageous,HENG MEI LENG YAN (flattened eyebrows and cold face): frown and look coldly, JI ZHI NU MU (point one’s fingers at somebody and stare at him with angry eyes): point and look at somebody furiously, CHEN MU QIE CHI (staring the eyes and gritting the teeth): staring and gritting with anger, FA ZHI ZHI LIE (with hair standing up and eye sockets tearing): boil with anger, LIU MEI DAO SHU (willow-leaf shaped eyebrows rose) : raise one’s eyebrows in anger... |
|
Anger as the agitation of qi inside the body |
QI DE TIAO JIAO (with so much qi that one stamps): stamp one’s feet with anger, FA PI QI (qi in the spleen exploded): lose temper, SHENG QI (generating qi): anger/get angry, SHENG MEN QI (generating silent qi): be in a sulk, NU QI (angry qi): anger/rage/fury, OU QI (be repressed with qi): sulk or repressed grievances, NU QI CHONG CHONG (angry qi rush out): huff and puff/in a fit of spleen/in a great rage/in a huff/seethe with anger, QI FEN TIAN YING (the breast is filled with angry qi) : be filled with indignation... |
|
Bodily sensations governed by interoceptive system |
|
|
Anger as the feeling of physical changes in the visceral organs |
NU CONG XIN TOU QI, E XIANG DAN BIAN SHENG (anger springs from the heart, and evil grows to the gall): be furious and nurse thoughts of revenge, DA DONG GAN HUO (violently stirred the liver fire): fly into a rage/ hit the roof), JI HUO GONG XIN (acute fire attacks the heart): burn with anger, FEI QI ZHA LE (the lungs exploded with qi): burst with rage... |
Comparable terms in English (see Table 4)
Table 4. English embodied words and idioms describing bodily states or sensations of anger
|
English Embodied words and idioms describing bodily states or sensations of anger |
|
|
Bodily states controlled by autonomic nervous system |
|
|
ANGER AS THE OUTPUT ENERGY ACCUMULATED IN THE BODY AS INTERNAL FLUID HEAT AND EVAPORATION PRESSURE |
heated, hot,slow burn,incensed,stew, blow up, fuming, inflame... E.g., Don't get hot under the collar. /Billy’s a hothead. /They were having a heated argument./ When the cop gave her a ticket, she got all hot and bothered./Don’t get a hernia! /When I found out, I almost burst a blood vessel. /He almost had a hemorrhage.
|
|
ANGER AS BODILY INJURY OR UNPLEASANT BODILY SENSATIONS |
cat fit, fit, rankling, inflamed, convulsed, exacerbated, nettled, chafed, sore/soreness,bitter...
|
|
ANGER AS REDNESS IN FACE AND NECK AREA |
She was scarlet with rage. /He got red with anger. /He was flushed with anger.
|
|
ANGER AS AGITATON |
She was shaking with anger./ I was hopping mad. /He was quivering with rage. He’s all worked up. She’s all wrought up.
|
|
ANGER IS INTERFERENCE WITH ACCURATE PERCEPTION |
She was blind with rage. /I was beginning to see red. /I was so mad I couldn’t see straight |
|
ANGER AS BREATH OR NOISE MADE BY BREATH |
huff/huffy, hissy, |
|
Bodily sensations of anger governed by interoceptive system |
|
|
ANGER AS PHYSIOLOGICAL SENSATION AND CHANGES IN THE VISCERAL ORGANS |
choler, gall, ill humor, choleric, galled, splenetic... |
Point 3:
Analysis
- The article seems to lead into the consideration that data will be quantified, but it is not the case. If it were done, the "pattern" would be more supported, since now the conclusions extracted from the analysis are not clearly justified.
Response 3:
Thank you for this insightful comment. We considered a more quantified approach to these data. However, we found this introduced specific issues, including a misleading sense of precision and rigor, since we cannot exhaustively collect all the words and idioms in one language and compare them with those in another language, not least because the number of the emotion words and idioms are not fixed but on the change with time. Furthermore, due to the diversity in conceptualization and associated conceptual systems of emotion, it is impossible to confirm one-to-one correspondence of the words and idioms across languages. Thus, we adopted more valid qualitative methods in the present manuscript.
Point 4:
Style
- Some sentences are too long and are not well connected between them.
- There are some minor mistakes (missing and unnecessary spaces)
- The numbers preceding some parts of the manuscript are not correct.
- Some of the references should be revise because book titles need to be in italics, for example.
Response 4:
We are grateful for the reviewer’s careful reading. We have shortened some long sentences and proofread the manuscript for several times and tried to correct some errors and typos in it.

Reviewer 2 Report
Abstract is too lengthy. Please revise it accordingly (200-250 words):
· An introductory statement which explains the background/significance of your review
· A brief description of your methodology and theoretical frameworks you work with
· A brief overview of the main contribution of your review
· A short concluding statement.
There is no Introduction. Please add it: Where you present the theme under investigation, the objectives and the questions you try to answer. You may add the outcomes of your review, presenting a clear contextual background. Please add the outline of your review: what parts it is comprised of, why, etc.
Conclusion: please critically reflect on how the literature enabled you to investigate this topic. Discuss shortcomings of reviewed sources, what should be done differently next time etc. You may want to make suggestions for future research.
Author Response
Cover Letter for revised Manuscript ID: brainsci-1738668:
Divergent conceptualization of embodied emotions in English and Chinese languages
Dear Editors for
Review #2
Point 1:
Comments and Suggestions for Authors
Abstract is too lengthy. Please revise it accordingly (200-250 words):
Response 1:
Many thanks for this helpful advice. We have shorten the abstract to 246 words as shown below:
Traditional cognitive linguistic theories acknowledge that human emotions are embodied, yet fail to distinguish dimensions that reflect the direction of neural signaling between brain and body. Differences exist across languages and cultures in whether embodied emotions are conceptualized as afferent (feelings from the body) or efferent (enacted through the body). This important distinction has been neglected in academic discourse, arguably as a consequence of the ‘lexical approach’, and the dominance within affective psychology of cognitive/semantic models that overlook the role of interoception as an essential component of affective experience. Empirical and theoretical advances in human neuroscience are driving a reappraisal of relationships between mind, brain and body, with particular relevance to emotions. Allostatic (predictive) control of internal bodily states is considered fundamental to the experience of emotions enacted through interoceptive sensory feelings, and through evoked physiological and physical actions mediated through efferent neural pathways. Embodied emotion concepts encompass these categorized outcomes of bidirectional brain-body interactions, yet can be differentiated further into afferent interoceptive and efferent/autonomic processes. Between languages, comparison of emotion words indicate the dominance of afferent/interoceptive processes in how embodied emotions are conceptualized in Chinese, while efferent autonomic processes feature more commonly in English. Correspondingly, in linguistic expressions of emotion, Chinese-speaking people are biased towards being more receptive, reflective and adaptive, whereas native English speakers may tend to be more reactive, proactive and interactive. Arguably, these distinct conceptual models of emotions may shape the perceived divergent values and ‘national character’ of Chinese and English–speaking cultures.
Point 2:
An introductory statement which explains the background/significance of your review. There is no Introduction. Please add it: Where you present the theme under investigation, the objectives and the questions you try to answer. You may add the outcomes of your review, presenting a clear contextual background. Please add the outline of your review: what parts it is comprised of, why, etc.
Response 2:
Thank you for the helpful advice. We have added two long paragraphs at the beginning of 1.0 on page 2, which reads as follows:
1.0. Re-examination of embodied emotional language within the framework of interoceptive cognitive theory
Reference to the body, is a feature shared across languages particularly when describing mental processes of emotion, reflecting the embodiment of emotional experience. Through the investigation of an impressive number and variety of languages, anthropological linguists claim that ‘emotions can be linguistically represented via literal somatic sensations (e.g., she blushed) or by body-part phrases referring to both literal and imaginary processes taking place inside or with the body (e.g., his hair stood on his head, his heart sank, it just makes my blood boil)’ [43] (p.148). Beyond this general tendency to link emotion states to bodily states, languages nevertheless vary dramatically in how they embody emotion concepts. Some researchers generalize three broad conceptualization of the mind and emotion categories globally as: (1) Abdominocentrism (i.e., the mind, including feeling, thinking and knowing, is conceptualized as being located in or around the abdomen region, e.g. in the belly, in the liver or in the kidney). Southern Asia, Polynesia and other disparate cultures, including Basque culture are the prototypical abdominocentric cultures. (2) Cardiocentrism (i.e., the mind and emotions are located in the heart region). Cardiocentrism is the traditional view of China, Korea, and Japan, with their shared similar philosophical and medical cultural models of holistic, heart-centering conceptualizations. (3) Cerebrocentrism (i.e., the mind is believed to be seated in the head or the brain region). This conception of the mind is mainly held by Greek-based West Asian, European, and North-African cultures, with the prototypical examples being the major Indo-European languages. [39, 48]
While some academics claim that many non-Western languages do not appear to differentiate between emotions and bodily sensations to the same extent that Western languages do [43], others propose that Western and non-Western languages are located at the two poles of a continuum between bodily transparency and cognitive granularity in the conceptualization of emotions. For example, Chinese is comparatively higher in bodily transparency but lower in cognitive granularity, while, Western languages manifest the opposite relationship [65].
Nonetheless, the term ‘embodied emotion’ remains a broad concept that encompasses the two dimensions of afferent (body to brain) versus efferent (brain to body) flow of information. Arguably, the contribution of the body to emotions should be differentiated along these complementary routes of brain-body interaction; i.e. the ‘bottom-up’ interoceptive sensory signals passing along the afferent pathways and the top-down autonomic nervous drive to change internal bodily state transmitted along the efferent pathway. A more granular understanding of embodied cognitive processes and the brain-body relationship will be enriched by consideration of this distinction.
Point 2:
- A brief description of your methodology and theoretical frameworks you work with·
Response 2:
Yes,we have added more details regarding the methodology for data-collection, stated as follows:
To explain further this point, the English words labeling the four so-called ‘basic’ emotions, namely, fear, anger, sadness and joy were collected from Roget’s 21st Century Thesaurus (the 7th edition) and on-line dictionary such as [26]. The Chinese words and idioms expressing the four ‘basic’ emotions and the bodily words that are associated with emotions were collected and selected from the Modern Chinese Dictionary [13] (7th Edition), Chinese Idioms Dictionary (2nd Edition)[64], Chinese Idiom Advance Dictionary [36], and from selected on-line dictionaries, specifically http://zhidao.baidu.com, http://xh.5156edu.com, http://chengyu.t086.com and http://www.hydcd.com. Identified emotion words and idioms were then compared between Chinese and English.
Point 3:
A brief overview of the main contribution of your review. A short concluding statement.
Conclusion: please critically reflect on how the literature enabled you to investigate this topic. Discuss shortcomings of reviewed sources, what should be done differently next time etc. You may want to make suggestions for future research.
Response 3:
Yes, thank you for this helpful advice. We have now added several sentences before the original conclusion, which now reads:
Conclusion
Traditional cognitive science applied to emotion has typically overlooked the role of interoception as the substrate for subjective awareness of emotion. Even models of embodied emotion have failed to differentiate the afferent from efferent pathways route of signaling between the body and the brain. Consequently, dominant cognitive linguistic theories also do not make such distinctions, and cannot account for variety of concepts across languages and cultures applied to the embodiment of emotions. Advances in affective psychology and interoceptive neuroscience, however, indicate that affective feelings emerge from interoceptive signals that are integrated and represented within specific interconnected brain regions (notably insula cortex) and are translated into subjective awareness and heuristic emotional concepts with varying degrees of bodily transparency and cognitive granularity, shaped by socio-cultural nuances [65]. Although in both Chinese and English languages, emotions are described with reference to bodily organs and physiological functions, we highlight how these languages differ with respect to the distinct emphasis on afferent versus efferent signaling in brain-body interaction, and argue that Chinese and English diverge in how emotion concepts are embodied. In Chinese, emotion concepts are more strongly linked to feelings arising from afferent representations of interoceptive states. However, in English, the overt physical and behavioural expressions of emotion including physiological reactions evoked by efferent autonomic pathways, are foregrounded. By extension, divergent conceptual systems of emotions within languages may foster distinct cultural attitudes and narratives regarding emotions: Chinese speakers, may be biased toward receptive, reflective and adaptive in how emotions impact lives and decisions, whereas English speakers, and others who share the same Western philosophical tradition, may be biased towards a more reactive and proactive stance in their affective and emotional behaviors.

Round 2
Reviewer 1 Report
The manuscript has significantly improved, but it still does not specify how the words or idioms listed for both Chinese and English were detected using thesauri.

Author Response
Cover Letter for revised Manuscript ID (round 2): brainsci-1738668:
Divergent conceptualization of embodied emotions in English and Chinese
languages
Dear Editors for
Review #1
The manuscript has significantly improved, but it still does not specify how the words or idioms listed for both Chinese and English were detected using thesauri.
Response:
We are indebted for this helpful comments. We have added more detailed descriptions to specify the methodologies of data collection from dictionaries and thesauri, which reads as follows (on page 4-5):
To explain further this point, the English words (synonyms) labeling the four so-called ‘basic’ emotions, namely, fear, anger, sadness and joy were collected from Roget’s 21st Century Thesaurus (the 7th edition) [26] as well as from on-line dictionary such as Collins Dictionary.com [6] and https://www.sketchengine.eu/skell [35], from which the emotion words with reference to body parts, physical states and physiological reactions were sorted out and illustrated with examples. As for the collection of Chinese data, we first searched the keywords ‘与人体内脏有关的成语’ (the idioms that are associated with visceral organs) and ‘描写愤怒/恐惧/悲伤/喜悦的词语/成语’(the words/idioms describing anger/fear/sadness/joy) on www.baidu.com,the most popular and frequently used search engine in China, from which several relevant websites were available like https://wenku.baidu.com/view/1bac147ee2bd960590c67796.html, http://zhidao.baidu.com, http://xh.5156edu.com, http://chengyu.t086.com and http://www.hydcd.com, wherein we singled out the embodied emotion words and idioms and subsumed them to the four basic emotions respectively. Then we consulted the definitions and their examples in The Modern Chinese Dictionary (7th Edition) [13] , Chinese Idioms Dictionary (2nd Edition) [65], Chinese Idiom Advance Dictionary [36] to make sure those embodied emotions words were fallen to the proper categories of emotion concepts. Furthermore, the embodied words and idioms used in the four great Chinese classics (i.e., A Dream in Red Mansions, The Romance of Three Kingdoms, Outlaws of Marshes, and Journey to the West.) were collected , from which those associated with emotions were picked out and compared with their English translations based on the Chinese-English parallel corpus of Shaoxing University of Arts and Sciences namely, Chinese-English parallel Corpus of A Dream of Red Mansions, Chinese-English parallel Corpus of The Romance of Three Kingdoms, Chinese-English parallel Corpus of Journey to the West, and Chinese-English parallel Corpus of Outlaws of Marshes (http://corpus.usx.edu.cn)[52] so that the embodied Chinese words and idioms could be compared with their English translations to hypothesize the rules underlying the conversion of emotion concepts across the two languages (e.g., whether or not the embodiment were maintained and which aspects of the embodiment were featured by the either language). The following are the findings from the collected data.

Reviewer 2 Report
I recommend to proof-read the text.
Author Response
Cover Letter for revised Manuscript ID (round 2): brainsci-1738668:
Divergent conceptualization of embodied emotions in English and Chinese
languages
Dear Editors for
Review #2
I recommend to proof-read the text.
Response:
Many thanks for the suggestion. We have proof-read the text twice and corrected several typos and small errors which are marked with blue highlight color in the resubmitted manuscript.
